# A *Vibrio* T6SS-Mediated Lethality in an Aquatic Animal Model

Hadar Cohen,[a] Chaya M. Fridman,[a] Motti Gerlic,[a] Dor Salomon[a]

ᵃDepartment of Clinical Microbiology and Immunology, Sackler Faculty of Medicine, Tel Aviv University, Tel Aviv, Israel

**ABSTRACT** Bacteria belonging to the genus *Vibrio* include many known and emerging pathogens. Horizontal gene transfer of pathogenicity islands is a major contributor to the emergence of new pathogenic *Vibrio* strains. Here, we use the brine shrimp *Artemia salina* as a model and show that the marine bacterium *Vibrio proteolyticus* uses a horizontally shared type VI secretion system, T6SS3, to intoxicate a eukaryotic host. Two T6SS3 effectors, which were previously shown to induce inflammasome-mediated pyroptotic cell death in mammalian phagocytic cells, contribute to this toxicity. Furthermore, we find a novel T6SS3 effector that also contributes to the lethality mediated by this system against *Artemia salina*. Therefore, our results reveal a T6SS that is shared among diverse vibrios and mediates host lethality, indicating that it can lead to the emergence of new pathogenic strains.

**IMPORTANCE** The rise in sea surface temperature has been linked to the spread of bacteria belonging to the genus *Vibrio* and the human illnesses associated with them. Since vibrios often share virulence traits horizontally, a better understanding of their virulence potential and determinants can prepare us for new emerging pathogens. In this work, we showed that a toxin delivery system found in various vibrios mediates lethality in an aquatic animal. Taken together with previous reports showing that the same system induces inflammasome-mediated cell death in mammalian phagocytic cells, our findings suggest that this delivery system and its associated toxins may contribute to the emergence of pathogenic strains.

**KEYWORDS** *Vibrio proteolyticus*, secretion, effector, virulence, *Artemia*, host-pathogen interaction, effector functions

Global warming and rising ocean water temperatures are linked to the spread of Gram-negative bacteria of the genus *Vibrio* and their associated human diseases (1). Since vibrios are known for their ability to horizontally acquire new virulence traits (2), their continuing spread requires an in-depth investigation of their pan-genome virulence potential.

*Vibrio proteolyticus* (*Vpr*) is a Gram-negative marine bacterium. *Vpr* strains were isolated from diseased corals (3) and were shown to be virulent to fish (4) and crustaceans (5). We previously reported that *Vpr* strain NBRC 13287 (ATCC 15338) has 3 anti-eukaryotic determinants: a secreted pore-forming hemolysin (VPRH) that kills mammalian cells, and 2 type VI secretion systems (T6SS1 and T6SS3) that deliver anti-eukaryotic effector proteins into mammalian phagocytic cells (6–8). The T6SS is a protein delivery apparatus widespread in Gram-negative bacteria (9). It delivers toxins, called effectors, directly into neighboring cells (10). Most T6SSs studied to date deliver antibacterial effectors into rival bacteria and thus play a role in interbacterial competitions (11). Few T6SSs were shown to deliver anti-eukaryotic effectors and thus play a role in virulence or defense against predation (10, 12). Here, we set out to investigate the virulence potential of these determinants in a widely used aquatic animal model, the saline lake-dwelling brine shrimp *Artemia salina*.

Challenging *Artemia* nauplii (larvae) with either wild-type or Δ*vprh Vpr* strains resulted in comparable high percentage of survival (Fig. 1A and Table S1 and S2), suggesting that VPRH does not play a significant role during the infection of this aquatic animal. To investigate the

Address correspondence to Motti Gerlic, mgerlic@tauex.tau.ac.il, or Dor Salomon, dorsalomon@mail.tau.ac.il.

The authors declare no conflict of interest.

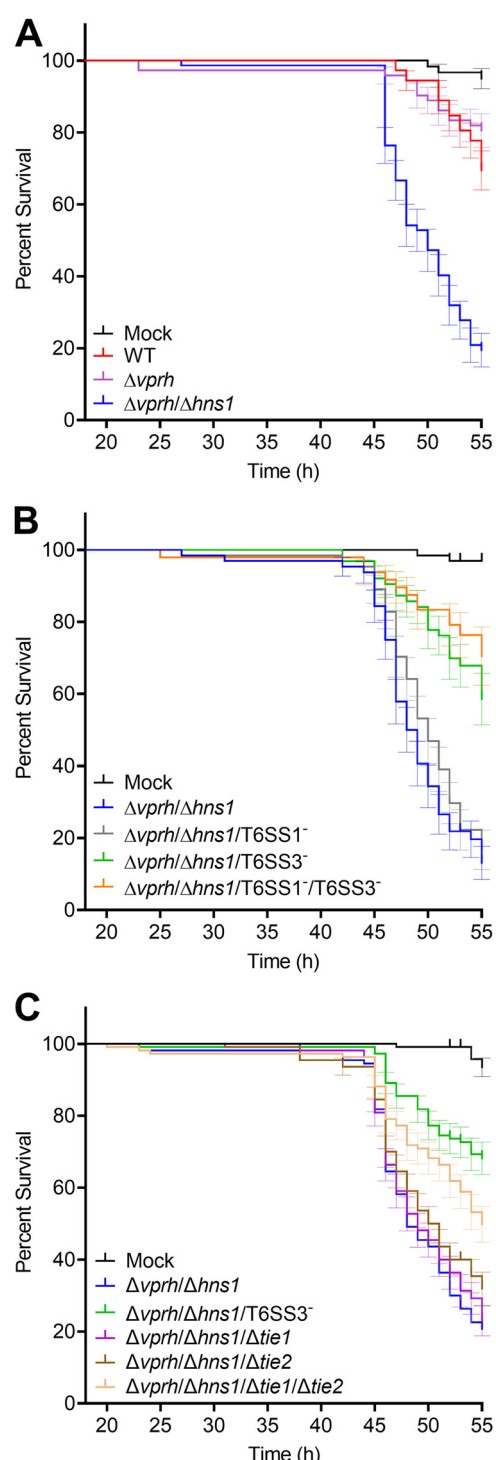

**FIG 1** *Vpr* T6SS3 mediates lethality in *Artemia* nauplii. (A to C) *Artemia* nauplii were challenged with the indicated *Vpr* strains, and survival was assessed 20 to 55 h postinfection (approximately $10^8$ bacteria were added to each well containing 2 nauplii). Data are shown as the mean ± SE of at least 3 biological replicates. WT, wild type.

virulence potential of the T6SSs, we used a strain in which we deleted *hns1*, encoding a histone-like nucleoid structuring protein that we previously showed represses the expression of T6SSS1 and T6SS3 (7). This Δ*vprh*/Δ*hns1* strain, which was shown to induce pyroptotic cell death in mammalian phagocytic cells (7), was significantly more lethal to the nauplii than the wild-type and Δ*vprh* strains (Fig. 1A and Table S1 and S2). Therefore, we posited that a *Vpr* T6SS is responsible for *Artemia* lethality.

To determine which T6SS is responsible for the *Vpr*-mediated lethality, we challenged *Artemia* with strains in which we inactivated either T6SS1, T6SS3, or both (Δ*vprh*/Δ*hns1*/T6SS1⁻, Δ*vprh*/Δ*hns1*/T6SS3⁻, or Δ*vprh*/Δ*hns1*/T6SS1⁻/T6SS3⁻, respectively) by deleting essential components of the systems (*tssG1* and *tssL3*, respectively). Inactivation of T6SS1 had no significant effect on *Artemia* survival (Fig. 1B and Table S3 and S4). However, the inactivation of T6SS3 (either alone or in combination with T6SS1) resulted in significantly higher survival of *Artemia* compared to the Δ*vprh*/Δ*hns1* strain (Fig. 1B and Table S3 and S4). Inactivation of T6SS2 by deletion of *vgrG2* had no effect on lethality (Fig. S1 and Table S5 and S6). Importantly, longer incubation times ($>$ 70 h) revealed that the wild-type *Vpr* is also lethal to *Artemia* at a later time point at which this lethality is largely T6SS3-dependent (Fig. S2 and Table S7 and S8). These results indicated that T6SS3, which we previously suggested to be horizontally shared between pathogenic vibrios and that induces pyroptotic cell death in mammalian phagocytic cells (7), mediates lethality in an aquatic animal.

We previously identified 2 *Vpr* T6SS3-secreted effectors, Tie1 and Tie2. We showed that these effectors are necessary and sufficient to induce pyroptotic cell death in mammalian phagocytic cells (7). Surprisingly, challenging *Artemia* nauplii with a strain in which we deleted both of these T6SS3 effectors (Δ*vprh*/Δ*hns1*/Δ*tie1*/Δ*tie2*) did not mirror the protection observed upon inactivation of T6SS3 (Fig. 1C and Table S9 and S10). Although it was less lethal than its parental Δ*vprh*/Δ*hns1* strain, Δ*vprh*/Δ*hns1*/Δ*tie1*/Δ*tie2* was still more lethal than the Δ*vprh*/Δ*hns1*/T6SS3⁻ strain. This result led us to hypothesize that *Vpr* delivers additional effectors contributing to the T6SS3-mediated lethality in *Artemia*.

To test our hypothesis, we performed a comparative proteomics analysis to reveal the T6SS3 secretome using strains over-expressing the predicted transcription regulator Ats3, which we previously identified as a T6SS3-specific activator (7), from a plasmid. We identified 5 proteins that were preferentially secreted by the wild-type (T6SS3⁺) strain compared to the T6SS3⁻ strain: 2 secreted structural components of T6SS3 (Hcp3 and VgrG3), 2 known secreted effectors (Tie1 and Tie2), and a hypothetical protein with no predicted activity or domain (WP_021706393.1) (Fig. 2A and Data set S1). We predicted that the latter is also a T6SS3 effector; following the previous nomenclature of *Vpr* T6SS3 effectors (7), we named it Tie3. A secretion assay using custom-made antibodies confirmed that Tie3 (predicted molecular weight of 33.6 kDa) is secreted in a T6SS3-dependent manner (Fig. 2B) and is not required for T6SS3 activity (Fig. S3). Although Tie3 is encoded outside the main *Vpr* T6SS3 gene cluster, it is found next to the virulence regulators ToxRS (13). Notably, its homologs are often encoded within or flanking a T6SS3-like gene cluster, or next to a known virulence factor (e.g., a VPRH homolog) (Fig. 2C).

The above results prompted us to investigate whether Tie3 plays a role in the T6SS3-mediated lethality in *Artemia*. Indeed, challenging *Artemia* with a strain in which all 3 effectors are deleted (Δ*vprh*/Δ*hns1*/Δ*tie1*/Δ*tie2*/Δ*tie3*) resulted in comparable survival to that observed for the Δ*vprh*/Δ*hns1*/T6SS3⁻ strain (Fig. 2D and Table S11 and S12). Notably, the deletion of *tie3* did not affect bacterial growth (Fig. S4). Importantly, introducing *tie3* into the *tie2* locus in the Δ*vprh*/Δ*hns1*/Δ*tie1*/Δ*tie2*/Δ*tie3* strain rescued the lethal phenotype (Fig. S5 and Table S13 and S14), indicating that the effect of *tie3* deletion on lethality was not a polar effect. Therefore, Tie3 is a novel effector that contributes to the T6SS3-mediated lethality in *Artemia*, together with the previously described effectors Tie1 and Tie2.

**Conclusions.** Here, we have revealed a role for a *Vibrio* T6SS during infection of an aquatic animal. We found that although the secreted hemolysin, VPRH, is lethal to mammalian cells, it does not appear to contribute to the virulence of *Vpr* in an aquatic animal model. Rather, T6SS3 is responsible for *Vpr*-mediated lethality in *Artemia*, using effectors that also induce pyroptotic cell death in mammalian phagocytic cells. In addition, we identified a novel T6SS effector, Tie3, that plays a role in T6SS3-mediated virulence; only deletion of all 3 known effectors, Tie1-3, recapitulated the loss of toxicity observed in the absence of a functional T6SS3. Notably, the function of these 3 effectors and their cellular targets inside the host remain unknown and require future investigation. Since this virulent T6SS was previously

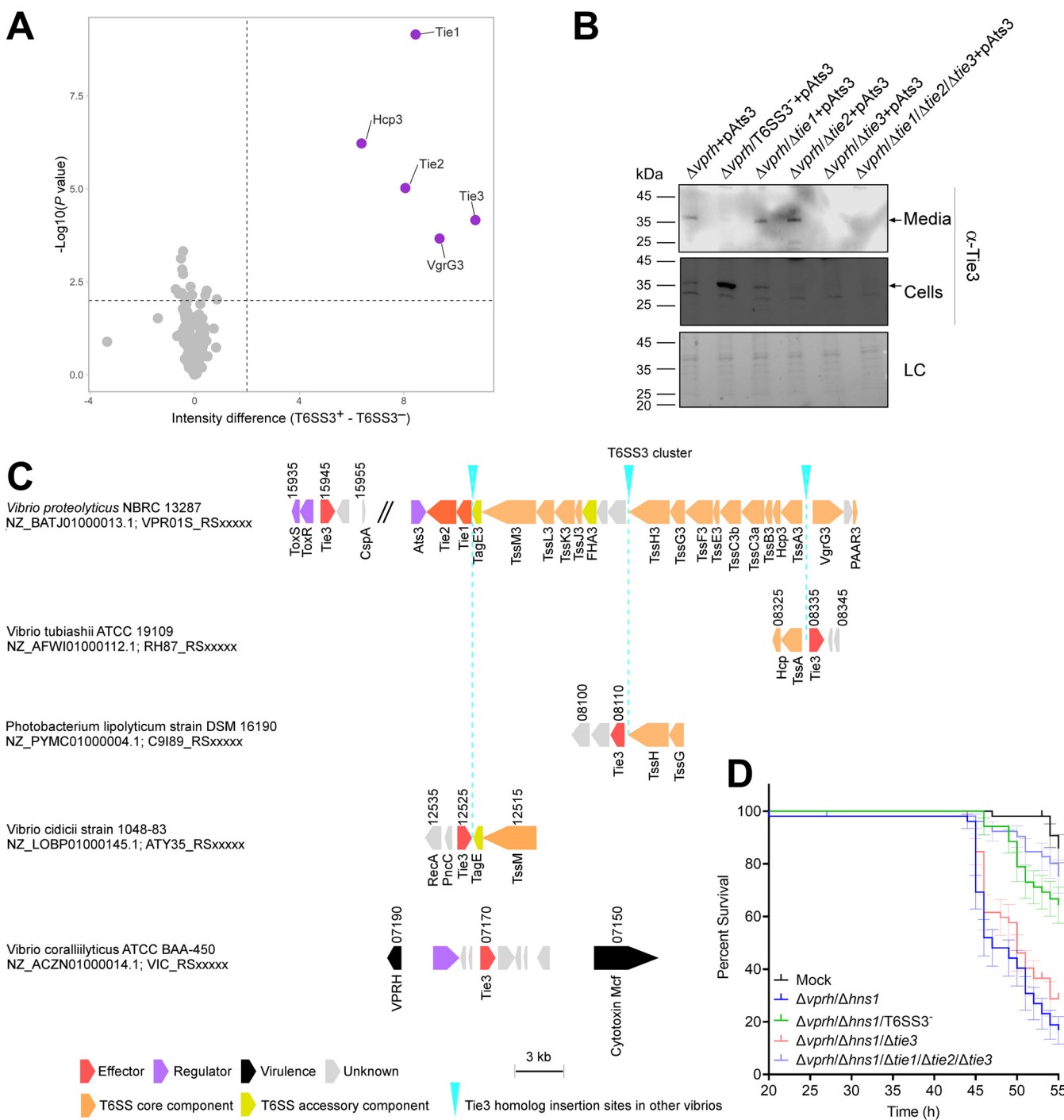

**FIG 2** Tie3 is a novel effector contributing to T6SS3-mediated lethality. (A) A volcano plot summarizing the comparative proteomics of proteins identified in the media of *Vpr* strains with an active (T6SS3$^+$) or inactive (T6SS3$^-$) T6SS3 and over-expressing Ats3 from a plasmid, using label-free quantification. The average difference in signal intensities between the T6SS3$^+$ and T6SS3$^-$ strains is plotted against the -Log$_{10}$ of Student's *t* test *P* values (*n* = 3 biological replicates). Proteins that were significantly more abundant in the secretome of the T6SS3$^+$ strain (difference in the average LFQ intensities > 2; *P* value < 0.03; with a minimum of 10 Razor unique peptides) are denoted in purple. (B) The expression (cells) and secretion (media) of Tie3 from the indicated strains containing a plasmid for the arabinose-inducible expression of Ats3 were detected by immunoblotting using specific antibodies against Tie3. Loading control (LC) is shown for total protein lysate. Arrows denote the expected size of Tie3 (33.6 kDa). The data are representative of 3 independent experiments with similar results. (C) *Vpr* T6SS3 gene cluster and genomic neighborhoods of selected Tie3 homologs. Genes are represented by arrows indicating the direction of transcription. Locus tags are denoted above; encoded proteins and known domains are denoted below. (D) *Artemia* nauplii were challenged with the indicated *Vpr* strains, and survival was assessed 20 to 55 h postinfection (approximately 10$^8$ bacteria were added to each well containing 2 nauplii). Data are shown as the mean ± SE of at least 3 independent experiments.

predicted to be horizontally shared between pathogenic vibrios and other marine bacteria (7), it may contribute to the emergence of new pathogenic strains.

## SUPPLEMENTAL MATERIAL

Supplemental material is available online only.
**SUPPLEMENTAL FILE 1**, XLSX file, 0.5 MB.
**SUPPLEMENTAL FILE 2**, PDF file, 0.5 MB.

## ACKNOWLEDGMENTS

We thank members of the Gerlic and Salomon labs for helpful discussions, and the Smoler Proteomics Center at the Technion for performing and analyzing the mass spectrometry data. The research was supported by the Recanati Foundation (to D.S.), the Margurt Shtultz grant (to M.G.), and the Israel Science Foundation (grant number 2174/22 to M.G.). C.M.F. was supported by a scholarship from the Clore Israel Foundation and a scholarship for outstanding doctoral students from the Orthodox community from the Council of Higher Education. This work was performed in partial fulfilment of the requirements for a Ph.D. degree for H.C. at the Sackler Faculty of Medicine, Tel Aviv University.

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
