## [Reviewer comments · Microbiology Spectrum]

Microbiology Spectrum

A *Vibrio* T6SS-mediated lethality in an aquatic animal model

Hadar Cohen, Chaya Fridman, Mordechai Gerlic, and Dor Salomon

Corresponding Author(s): Dor Salomon, Tel Aviv University

Review Timeline:

Submission Date:	March 13, 2023
Editorial Decision:	April 1, 2023
Revision Received:	May 11, 2023
Editorial Decision:	May 29, 2023
Revision Received:	May 29, 2023
Accepted:	May 31, 2023

Editor: Eric Cascales

Reviewer(s): Disclosure of reviewer identity is with reference to reviewer comments included in decision letter(s). The following individuals involved in review of your submission have agreed to reveal their identity: Marylise Duperthuy (Reviewer #1)

Transaction Report:

DOI: <https://doi.org/10.1128/spectrum.01093-23>

April 1, 2023

Dr. Dor Salomon
Tel Aviv University
Clinical Microbiology and Immunology
Tel Aviv
Israel

Re: Spectrum01093-23 (A *Vibrio* T6SS-mediated lethality in a marine animal model)

Dear Dor,

Thank you for submitting your manuscript to Microbiology Spectrum. Your manuscript has been sent to three external experts. As you will see in their comments pasted below, they all acknowledge that you report important findings but they also require additional experimentations and statistical analyses, notably complementation experiments, virulence assays in the wild-type background. The reviewers also suggest modifications in the text, e.g., notably a more detailed description of *Vibrio* parahaemolyticus in the introduction section and more description of some methods. When submitting the revised version of your paper, please provide (1) point-by-point responses to the issues raised by the reviewers as file type "Response to Reviewers," not in your cover letter, and (2) a PDF file that indicates the changes from the original submission (by highlighting or underlining the changes) as file type "Marked Up Manuscript - For Review Only". Please use this link to submit your revised manuscript - we strongly recommend that you submit your paper within the next 60 days or reach out to me. Detailed instructions on submitting your revised paper are below.

Link Not Available

Sincerely,
Eric

Eric Cascales

Journals Department
Reviewer comments:

Reviewer #1 (Comments for the Author):

This study focusses on the role of the T6SS in the virulence of *V. proteolyticus*. The authors used an *Artemia* infection model to determine the role of the T6SS3 in the virulence of *Vpr* and identified a new potential virulence effector secreted through this system. Overall, although short, the manuscript is well written, but some experiments and statistics are lacking to support the conclusions. Besides, the manuscript is lacking a thorough discussion of the results. Please find below my comments:

- 1) Are *Artemia salina* and *Artemia nauplii* the same animal? The authors seem to use both names interchangeably.
- 2) The authors tested mutants of the T6SS1 and the T6SS3 and show that the T6SS3 is important for the virulence of Vpr in an *Artemia* model. It would have been interesting to test the T6SS2 too. Is there any reason why it was not done?
- 3) The authors observe a strong attenuation of the virulence upon T6SS3 deletion in a *hns* mutant background. However, there are no evidence that the T6SS3 is important for the virulence of Vpr in a WT background or that the T6SS3 is expressed *in vivo* during the infection. Thus, it would be important to test the importance of the T6SS3 in the virulence of Vpr by using a T6SS3 mutant in a WT background. Similar observation can be made for the Tie3 effector.
- 4) In figure 2C, it appears that the Tie1/2/3 mutant is less virulent than the T6SS3 mutant, a difference that seems significant. In fact, some statistics between the strains are missing (only the comparison with the WT or the *hns* mutant are presented in Fig 1 and 2). This observation is somehow puzzling considering that the Tie effectors are all secreted through the T6SS3. This result should be discussed.
- 5) None of the results regarding the mutants have been validated by the complementation of the mutations, so it cannot be excluded that the results obtained are not due to a polar effect of the mutations, especially regarding Tie3 which is located next to ToxRS, a system well known for its role in virulence in *Vibrio*.

Reviewer #2 (Comments for the Author):

The observation presented here provides evidence that the T6SS-3 of *Vibrio proteolyticus* contributes to the pathogenic potential of the bacterium towards a shrimp animal model. The authors also identify a novel T6SS-3 effector that contributes to the observed phenotype. Data presented in this paper is convincing and the manuscript is well written. Please find below my comments:

- There is no information about *Vibrio proteolyticus* or the T6SS in the introductory paragraph. A few sentences to describe the importance of the bacterium (which hosts does it infect? does it have a large impact on animal populations?) and the T6SS, as well as some basic information about how the T6SS functions, are required.
- On line 27, it is mentioned that the toxin delivery system is found on mobile genomic islands. However, the manuscript does not provide any evidence for this statement. Therefore, this statement should be removed.
- One of the main findings of this study is that Tie3 is a novel effector secreted by the T6SS-3. Data from Supplementary Fig. 1 provides compelling evidence for this finding. Therefore, Supplementary Fig. 1 should be part of the main manuscript. I suggest that only the α -Tie3 part of the figure is necessary to be included in the main text (along with the loading control); the α -Hcp3 part of the figure can be kept in the supplementary material. Also, please include the Tie3 protein size in kDa in the main text and in the figure legend.
- On line 45, please clarify that the term "nauplii" refers to shrimp larvae.
- On line 49, the introduction of the *hns* mutant is surprising because it has not been mentioned before. Please include a sentence prior to the one started on line 47 ("To investigate the virulence potential...") to describe that *Hns* is a previously identified negative regulator of the T6SS. Similarly, please describe the *Ats3* activator prior to its introduction on line 75.
- On line 57, it should be clarified which essential components of the systems were deleted.
- Please include a statement about Tie3 in the conclusion.
- The supplementary file mentions that the proteomics data has been deposited in the ProteomeXchange Consortium dataset. How can readers access this data? An accession/project number needs to be included in the methods.

Reviewer #3 (Comments for the Author):

This work summarizes a new finding that *V. proteolyticus* requires the T6SS3 and a novel effector to cause disease in *Artemia* shrimp. Overall, I found the paper to be clearly written and the data appear to support the conclusions. I was a bit confused about the methods used for the *Artemia* infection assays, but these could easily be addressed by adding some additional detail to the supplemental methods. Below are my suggestions for the authors to consider.

1. I realize this is an observation with strict word limits, but given the emphasis on vibrios as pathogens in the intro, it may be helpful to mention whether *Vibrio proteolyticus* is a natural pathogen in the marine environment, and/or if it is viewed as a possible emerging pathogen for marine animals or humans. It is my understanding that *Artemia* is not a marine shrimp, so the infection assays are a measurement of what could be, not necessarily a characterization of a natural host-pathogen interaction. I think this is completely fine, but clarification for the reader's sake could be considered.
2. The supplemental methods say the *Artemia* "were scored" at indicated time points and the supplemental tables note the metric reported is "Subjects at risk" while the main figure graphs show "Probability of survival" and state that "survival was assessed" in the legend. It is unclear to me what is being evaluated to assess "risk" or "survival" and how these terms might be different. Is it just the number of individuals that are alive at a given time point, based on movement or response to stimulus?

Could the authors elaborate on how Artemia were scored in the supplemental methods?

3. I really appreciate the authors including the raw data for the infection assays as supplemental tables but was confused about how those data are transformed to what is shown in the main figures. For example, Fig 1B shows the probability of survival for mock treatment being very near 100 but the data shown in table S2 shows the number of surviving Artemia going from 64 to 31, which is 48% survival, if the numbers in the table are indicating surviving individuals. Because the trends in the supplemental tables reflect what is shown in the main figures, the author's conclusions seem to hold, but it would be helpful to provide clarification regarding how the "probability of survival" was calculated from the values shown in the supplemental tables.

4. What do the authors mean by "median survival" in the figures and why was it undefined for some of the treatments?

5. Could the authors please explain in the supplemental methods how the hazard ratio was calculated.

Staff Comments:

Preparing Revision Guidelines

Please return the manuscript within 60 days; if you cannot complete the modification within this time period, please contact me. If you do not wish to modify the manuscript and prefer to submit it to another journal, please notify me of your decision immediately so that the manuscript may be formally withdrawn from consideration by Microbiology Spectrum.

This study focusses on the role of the T6SS in the virulence of *V. proteolyticus*. The authors used an *Artemia* infection model to determine the role of the T6SS3 in the virulence of *Vpr* and identified a new potential virulence effector secreted through this system. Overall, although short, the manuscript is well written, but some experiments and statistics are lacking to support the conclusions. Besides, the manuscript is lacking a thorough discussion of the results. Please find below my comments:

1) Are *Artemia salina* and *Artemia nauplii* the same animal? The authors seem to use both names interchangeably.

2) The authors tested mutants of the T6SS1 and the T6SS3 and show that the T6SS3 is important for the virulence of *Vpr* in an *Artemia* model. It would have been interesting to test the T6SS2 too. Is there any reason why it was not done?

3) The authors observe a strong attenuation of the virulence upon T6SS3 deletion in a *hns* mutant background. However, there are no evidence that the T6SS3 is important for the virulence of *Vpr* in a WT background or that the T6SS3 is expressed in vivo during the infection. Thus, it would be important to test the importance of the T6SS3 in the virulence of *Vpr* by using a T6SS3 mutant in a WT background. Similar observation can be made for the Tie3 effector.

4) In figure 2C, it appears that the Tie1/2/3 mutant is less virulent than the T6SS3 mutant, a difference that seems significant. In fact, some statistics between the strains are missing (only the comparison with the WT or the *hns* mutant are presented in Fig 1 and 2). This observation is somehow puzzling considering that the Tie effectors are all secreted through the T6SS3. This result should be discussed.

5) None of the results regarding the mutants have been validated by the complementation of the mutations, so it cannot be excluded that the results obtained are not due to a polar effect of the mutations, especially regarding Tie3, which is located next to ToxRS, a system well known for its role in virulence in *Vibrio*.

The observation presented here provides evidence that the T6SS-3 of *Vibrio proteolyticus* contributes to the pathogenic potential of the bacterium towards a shrimp animal model. The authors also identify a novel T6SS-3 effector that contributes to the observed phenotype. Data presented in this paper is convincing and the manuscript is well written. Please find below my comments:

- There is no information about *Vibrio proteolyticus* or the T6SS in the introductory paragraph. A few sentences to describe the importance of the bacterium (which hosts does it infect? does it have a large impact on animal populations?) and the T6SS, as well as some basic information about how the T6SS functions, are required.
- On line 27, it is mentioned that the toxin delivery system is found on mobile genomic islands. However, the manuscript does not provide any evidence for this statement. Therefore, this statement should be removed.
- One of the main findings of this study is that Tie3 is a novel effector secreted by the T6SS-3. Data from Supplementary Fig. 1 provides compelling evidence for this finding. Therefore, Supplementary Fig. 1 should be part of the main manuscript. I suggest that only the α -Tie3 part of the figure is necessary to be included in the main text (along with the loading control); the α -Hcp3 part of the figure can be kept in the supplementary material. Also, please include the Tie3 protein size in kDa in the main text and in the figure legend.
- On line 45, please clarify that the term “nauplii” refers to shrimp larvae.
- On line 49, the introduction of the *hns* mutant is surprising because it has not been mentioned before. Please include a sentence prior to the one started on line 47 (“To investigate the virulence potential...”) to describe that Hns is a previously identified negative regulator of the T6SS. Similarly, please describe the Ats3 activator prior to its introduction on line 75.
- On line 57, it should be clarified which essential components of the systems were deleted.
- Please include a statement about Tie3 in the conclusion.
- The supplementary file mentions that the proteomics data has been deposited in the ProteomeXchange Consortium dataset. How can readers access this data? An accession/project number needs to be included in the methods.

A point-by-point reply to reviewers' comments

Thank you for submitting your manuscript to Microbiology Spectrum. Your manuscript has been sent to three external experts. As you will see in their comments pasted below, they all acknowledge that you report important findings but they also requires additional experimentations and statistical analyses, notably complementation experiments, virulence assays in the wild-type background, . The reviewers also suggest modifications in the text, e.g., notably a more detailed description of *Vibrio parahaemolyticus* in the introduction section and more description of some methods. When submitting the revised version of your paper, please provide (1) point-by-point responses to the issues raised by the reviewers as file type "Response to Reviewers," not in your cover letter, and (2) a PDF file that indicates the changes from the original submission (by highlighting or underlining the changes) as file type "Marked Up Manuscript - For Review Only". Please use this link to submit your revised manuscript - we strongly recommend that you submit your paper within the next 60 days or reach out to me. Detailed instructions on submitting your revised paper are below.

We thank the reviewers for their helpful comments and suggestions. As detailed below, we addressed all the suggestions with text changes and additional experiments. We also now refer to *Artemia* as an aquatic animal rather than a marine animal, since it dwells in lakes rather than in the ocean.

Reviewer #1 (Comments for the Author):

This study focusses on the role of the T6SS in the virulence of *V. proteolyticus*. The authors used an *Artemia* infection model to determine the role of the T6SS3 in the virulence of Vpr and identified a new potential virulence effector secreted through this system. Overall, although short, the manuscript is well written, but some experiments and statistics are lacking to support the conclusions. Besides, the manuscript is lacking a thorough discussion of the results. Please find below my comments:

We thank the reviewer for their suggestions. As described below, the revised version includes additional experiments to address the comments. We also expanded our discussion (within the 1,200-word limit of the Observation format).

1) Are *Artemia salina* and *Artemia nauplii* the same animal? The authors seem to use both names interchangeably.

Nauplii are the first larval stage of crustaceans. This is now mentioned after the first appearance of the term (line 52).

2) The authors tested mutants of the T6SS1 and the T6SS3 and show that the T6SS3 is important for the virulence of Vpr in an *Artemia* model. It would have been interesting to test the T6SS2 too. Is there any reason why it was not done?

We now include Supplementary Fig. S1, showing that inactivation of T6SS2 does not affect the toxicity towards *Artemia*. See also accompanying text (lines 68-69). T6SS2 was not tested in the original manuscript since we did not find it to be active in previous works (Ray et al, *mBio*, 2016 and Cohen et al., *eLife*, 2022).

3) The authors observe a strong attenuation of the virulence upon T6SS3 deletion in a *hns* mutant background. However, there are no evidence that the T6SS3 is important for the virulence of Vpr in a WT background or that the T6SS3 is expressed *in vivo* during the infection. Thus, it would be important to test the importance of the T6SS3 in the virulence of Vpr by using a T6SS3 mutant in a WT background. Similar observation can be made for the Tie3 effector.

Following the reviewer's suggestion, we now show that T6SS3 contributes to lethality mediated by the WT strain (Supplementary Fig. S2). Please note that the toxicity of the WT strain is mild and takes longer to manifest compared to that of the $\Delta hns1$ strain (in which T6SS3 is hyper-activated). Therefore, we did not investigate the contribution of Tie3 to lethality in the WT background. Please also see the accompanying text in lines 69-71.

4) In figure 2C, it appears that the Tie1/2/3 mutant is less virulent than the T6SS3 mutant, a

difference that seems significant. In fact, some statistics between the strains are missing (only the comparison with the WT or the *hns* mutant are presented in Fig 1 and 2). This observation is somehow puzzling considering that the Tie effectors are all secreted through the T6SS3. This result should be discussed.

Our statistical analysis indicates no significant difference between the Hazard ratio of the *tie1-3* mutant and the T6SS3 mutant in this figure (P value = 0.162). Moreover, please note that there is also no difference between these two strains in a new set of experiments performed as part of the revision (Fig. S5; see below). Statistical analyses for comparisons between all strains are now provided as supplementary Tables for all survival plots.

5) None of the results regarding the mutants have been validated by the complementation of the mutations, so it cannot be excluded that the results obtained are not due to a polar effect of the mutations, especially regarding Tie3 which is located next to ToxRS, a system well known for its role in virulence in *Vibrio*.

We now show that introducing *tie3* into the *tie2* locus (at the end of the T6SS3 gene cluster so that it remains under T6SS3 regulation) in a *tie1-3* mutant rescues the loss of toxicity (Fig. S5). These results indicate that the loss of toxicity upon *tie3* deletion was not due to a polar effect. The complementation of Tie1 and Tie2 was previously reported (Cohen et al., *eLife*, 2022), albeit for toxicity in murine macrophages rather than *Artemia*. Please also see the accompanying text (lines 103-106).

Reviewer #2 (Comments for the Author):

The observation presented here provides evidence that the T6SS-3 of *Vibrio proteolyticus* contributes to the pathogenic potential of the bacterium towards a shrimp animal model. The authors also identify a novel T6SS-3 effector that contributes to the observed phenotype. Data presented in this paper is convincing and the manuscript is well written. Please find below my comments:

- There is no information about *Vibrio proteolyticus* or the T6SS in the introductory paragraph. A few sentences to describe the importance of the bacterium (which hosts does it infect? does it have a large impact on animal populations?) and the T6SS, as well as some basic information about how the T6SS functions, are required.

As suggested, we included additional information on *V. proteolyticus* and the T6SS in the second paragraph.

- On line 27, it is mentioned that the toxin delivery system is found on mobile genomic islands. However, the manuscript does not provide any evidence for this statement. Therefore, this statement should be removed.

The sentence was revised accordingly.

- One of the main findings of this study is that Tie3 is a novel effector secreted by the T6SS-3. Data from Supplementary Fig. 1 provides compelling evidence for this finding. Therefore, Supplementary Fig. 1 should be part of the main manuscript. I suggest that only the α -Tie3 part of the figure is necessary to be included in the main text (along with the loading control); the α -Hcp3 part of the figure can be kept in the supplementary material. Also, please include the Tie3 protein size in kDa in the main text and in the figure legend.

The figures were rearranged as suggested, and the predicted MW of Tie3 is now noted in the text and in the figure legend.

- On line 45, please clarify that the term "nauplii" refers to shrimp larvae.

A clarification was added.

- On line 49, the introduction of the hns mutant is surprising because it has not been mentioned before. Please include a sentence prior to the one started on line 47 ("To investigate the

virulence potential...") to describe that Hns is a previously identified negative regulator of the T6SS. Similarly, please describe the Ats3 activator prior to its introduction on line 75.

The suggested information was added to the text (lines 56-57 and 85-86).

- On line 57, it should be clarified which essential components of the systems were deleted.

The information was added (tssG1 for T6SS1, tssL3 for T6SS3, and vgrG2 for the newly added T6SS2). See lines 64-65 and 68.

- Please include a statement about Tie3 in the conclusion.

We expanded the Conclusions section, which now includes more information on Tie3.

- The supplementary file mentions that the proteomics data has been deposited in the ProteomeXchange Consortium dataset. How can readers access this data? An accession/project number needs to be included in the methods.

Information on the publicly available proteomics raw data was added to the relevant Methods section.

Reviewer #3 (Comments for the Author):

This work summarizes a new finding that *V. proteolyticus* requires the T6SS3 and a novel effector to cause disease in *Artemia* shrimp. Overall, I found the paper to be clearly written and the data appear to support the conclusions. I was a bit confused about the methods used for the *Artemia* infection assays, but these could easily be addressed by adding some additional detail to the supplemental methods. Below are my suggestions for the authors to consider.

1. I realize this is an observation with strict word limits, but given the emphasis on vibrios as pathogens in the intro, it may be helpful to mention whether *Vibrio proteolyticus* is a natural pathogen in the marine environment, and/or if it is viewed as a possible emerging pathogen for marine animals or humans. It is my understanding that *Artemia* is not a marine shrimp, so the infection assays are a measurement of what could be, not necessarily a characterization of a natural host-pathogen interaction. I think this is completely fine, but clarification for the reader's sake could be considered.

As suggested, additional information was added to the second introductory paragraph.

2. The supplemental methods say the *Artemia* "were scored" at indicated time points and the supplemental tables note the metric reported is "Subjects at risk" while the main figure graphs show "Probability of survival" and state that "survival was assessed" in the legend. It is unclear to me what is being evaluated to assess "risk" or "survival" and how these terms might be different. Is it just the number of individuals that are alive at a given time point, based on movement or response to stimulus? Could the authors elaborate on how *Artemia* were scored in the supplemental methods?

We thank the reviewer for this comment, and we agree that the terminology was confusing and lacked appropriate explanations. For clarity, we changed "probability of survival" to "percent survival" as the units for the survival plots Y axes. Moreover, for simplicity, we also removed the text referring to hazard ratios from the description of the results. The terms "subjects at risk", "median survival time", and "hazard ratio" are now explained in the supplementary methods section, and the relevant data are provided in supplementary tables:

"Subjects at risk" refers to the viable *Artemia* included in ongoing experiments at the indicated timepoint. They are not survival rates as shown in the survival plots, since the plots show data from several independent experiments, not all of which ended at the same timepoint.

"Percent survival" was calculated as surviving subjects out of the subjects at risk for each time point.

"Median survival time" is the time at which only 50% of the *Artemia* population was still alive. If more than 50% of the population was alive at the end of the experiment, then the value was undefined.

"Hazard ratio" is the ratio between the hazard of two treatments. The Hazard is essentially the rate of mortality or "slope" of the survival curve.

We now also explain how Artemia viability was scored in the supplementary methods (the lack of movement for 10 seconds was considered as “non-viable”).

3. I really appreciate the authors including the raw data for the infection assays as supplemental tables but was confused about how those data are transformed to what is shown in the main figures. For example, Fig 1B shows the probability of survival for mock treatment being very near 100 but the data shown in table S2 shows the number of surviving Artemia going from 64 to 31, which is 48% survival, if the numbers in the table are indicating surviving individuals. Because the trends in the supplemental tables reflect what is shown in the main figures, the author's conclusions seem to hold, but it would be helpful to provide clarification regarding how the "probability of survival" was calculated from the values shown in the supplemental tables.

Please see our reply to comment #2.

4. What do the authors mean by "median survival" in the figures and why was it undefined for some of the treatments?

Please see our reply to comment #2.

5. Could the authors please explain in the supplementary methods how the hazard ratio was calculated.

Please see our reply to comment #2.

May 29, 2023

Dr. Dor Salomon
Tel Aviv University
Clinical Microbiology and Immunology
Tel Aviv
Israel

Re: Spectrum01093-23R1 (A *Vibrio* T6SS-mediated lethality in an aquatic animal model)

Dear Dor:

Thank you for submitting your revised manuscript to Microbiology Spectrum. It was sent back to the three original reviewers who acknowledge that you satisfactorily addressed their comments and recommend publication. Reviewer #2 is however suggesting very minor text changes, and I will proceed to final acceptance as soon as I receive the revised manuscript with these few modification. As these revisions are quite minor, I expect that you should be able to turn in the revised paper in less than 30 days, if not sooner.

When submitting the revised version of your paper, please provide (1) point-by-point responses to the issues raised by the reviewers as file type "Response to Reviewers," not in your cover letter, and (2) a PDF file that indicates the changes from the original submission (by highlighting or underlining the changes) as file type "Marked Up Manuscript - For Review Only". Please use this link to submit your revised manuscript. Detailed instructions on submitting your revised paper are below.

Link Not Available

Sincerely,
Eric

Eric Cascales

Reviewer comments:

Reviewer #1 (Comments for the Author):

The authors made substantial revisions, including new data. The results of this study are convincing. I am satisfied with their answers to my former questions.

Reviewer #2 (Comments for the Author):

I thank the authors for addressing all my major concerns. My final minor recommendations are:

On line 51, please change "lakes-dwelling" to "lake-dwelling"

On line 70, please change "at later timepoint" to "at a later time point"

On line 71, please change "and that" to "and which"

On line 116, please delete "the" from "in the T6SS3-mediated virulence"

Reviewer #3 (Comments for the Author):

The authors have addressed all of my comments. I have no further suggestions.

Preparing Revision Guidelines

Please return the manuscript within 60 days; if you cannot complete the modification within this time period, please contact me. If you do not wish to modify the manuscript and prefer to submit it to another journal, please notify me of your decision immediately so that the manuscript may be formally withdrawn from consideration by Microbiology Spectrum.

I thank the authors for addressing all my major concerns. My final minor recommendations are:

On line 51, please change "lakes-dwelling" to "lake-dwelling"

On line 70, please change "at later timepoint" to "at a later time point"

On line 71, please change "and that" to "and which"

On line 116, please delete "the" from "in the T6SS3-mediated virulence"

A point-by-point reply to reviewers' comments

We again thank the reviewers for their helpful comments and suggestions.

Reviewer #2 (Comments for the Author):

I thank the authors for addressing all my major concerns. My final minor recommendations are:

On line 51, please change "lakes-dwelling" to "lake-dwelling"

The text was revised accordingly.

On line 70, please change "at later timepoint" to "at a later time point"

The text was revised accordingly.

On line 71, please change "and that" to "and which"

The text was revised to "at which".

On line 116, please delete "the" from "in the T6SS3-mediated virulence"

The text was revised accordingly.

May 31, 2023

Dr. Dor Salomon
Tel Aviv University
Clinical Microbiology and Immunology
Tel Aviv
Israel

Re: Spectrum01093-23R2 (A *Vibrio* T6SS-mediated lethality in an aquatic animal model)

Dear Dor:

Thank you for incorporating the minor text changes. I am please to accept your manuscript for publication, and I am forwarding it to the ASM Journals Department. You will be notified when your proofs are ready to be viewed.

Sincerely,
Eric

Eric Cascales
Editor, Microbiology Spectrum
